# Genome-Wide Analysis of the *SRPP*/*REF* Gene Family in *Taraxacum kok-saghyz* Provides Insights into Its Expression Patterns in Response to Ethylene and Methyl Jasmonate Treatments

**DOI:** 10.3390/ijms25136864

**Published:** 2024-06-22

**Authors:** Huan He, Jiayin Wang, Zhuang Meng, Paul P. Dijkwel, Pingping Du, Shandang Shi, Yuxuan Dong, Hongbin Li, Quanliang Xie

**Affiliations:** 1Key Laboratory of Xinjiang Phytomedicine Resource and Utilization of Ministry of Education, Xinjiang Production and Construction Corps Key Laboratory of Oasis Town and Mountain-basin System Ecology, College of Life Sciences, Shihezi University, Shihezi 832003, China; he_huan026@163.com (H.H.); wjyinee@163.com (J.W.); zhuangmeng610@163.com (Z.M.); dopingping@126.com (P.D.); shi_shandang@163.com (S.S.); dyx501006130@163.com (Y.D.); 2School of Natural Sciences, Massey University, Tennent Drive, Palmerston North 4474, New Zealand; p.dijkwel@massey.ac.nz

**Keywords:** *Taraxacum kok-saghyz*, *TkSRPP*/*REFs*, natural rubber, genetic evolution, ethylene, methyl jasmonate, functional analysis

## Abstract

*Taraxacum kok-saghyz* (TKS) is a model plant and a potential rubber-producing crop for the study of natural rubber (NR) biosynthesis. The precise analysis of the NR biosynthesis mechanism is an important theoretical basis for improving rubber yield. The small rubber particle protein (SRPP) and rubber elongation factor (REF) are located in the membrane of rubber particles and play crucial roles in rubber biosynthesis. However, the specific functions of the *SRPP*/*REF* gene family in the rubber biosynthesis mechanism have not been fully resolved. In this study, we performed a genome-wide identification of the 10 *TkSRPP* and 2 *TkREF* genes’ family members of Russian dandelion and a comprehensive investigation on the evolution of the ethylene/methyl jasmonate-induced expression of the SRPP/REF gene family in TKS. Based on phylogenetic analysis, 12 TkSRPP/REFs proteins were divided into five subclades. Our study revealed one functional domain and 10 motifs in these proteins. The *SRPP*/*REF* protein sequences all contain typical REF structural domains and belong to the same superfamily. Members of this family are most closely related to the orthologous species *T. mongolicum* and share the same distribution pattern of *SRPP*/*REF* genes in *T. mongolicum* and *L. sativa*, both of which belong to the family Asteraceae. Collinearity analysis showed that segmental duplication events played a key role in the expansion of the *TkSRPP/REFs* gene family. The expression levels of most *TkSRPP*/*REF* members were significantly increased in different tissues of *T. kok-saghyz* after induction with ethylene and methyl jasmonate. These results will provide a theoretical basis for the selection of candidate genes for the molecular breeding of *T. kok-saghyz* and the precise resolution of the mechanism of natural rubber production.

## 1. Introduction

Natural rubber (NR) is mainly composed of cis-1,4-polyisoprene polymers, which are indispensable and strategically valuable biologically scarce materials in the fields of healthcare and construction due to their strong impact resistance, abrasion resistance, high flexibility, and superior resilience properties [1,2]. More than 2500 species of plants can produce NR, but only some can produce high molecular-weight rubber as a high-performance raw material for commercial needs [3,4,5,6]. Among these, the *Hevea brasiliensis* [7], *Parthenium argentatum* [8], *Lactuca sativa*, and the genus Russian dandelion produce natural rubber with an average molecular weight of up to 1000 kg/mol [2]. Currently, *H. brasiliensis* is the only source of NR for the manufacturing industry and cannot be replaced with synthetic materials. However, due to its special growing conditions, limited output, increasing demand, pathogens, and allergic reactions [9,10], it is urgent to explore alternative plant sources of natural rubber.

*Taraxacum kok-saghyz* L. Rodin (TKS) is a perennial herbaceous plant with high NR yield, good quality, low cost, and it can be used as a candidate alternative crop to natural rubber [11,12]. Compared to other rubber-producing plants such as the annual herbaceous plant *P. argentatum* [8], TKS produces NR with higher molecular weights, increased cis-1,4-polyisoprene content, and valuable by-products including inulin and bioethanol [13]. *T. kok-saghyz* had five times the NR yield compared to dandelion (*Taraxacum brevicorniculatum*, TB) [1]. However, the molecular mechanism of natural rubber biosynthesis (NRB) in TKS remains incomplete and elusive.

The precise analysis of the molecular regulation mechanism of natural rubber biosynthesis is important for the development of the natural rubber industry. From previous studies, it is known that *T. kok-saghyz* use both the Mevalonate pathway (MVA) and Methylerythritol pathway (MEP) to synthesize natural rubber, and 102 genes were identified in the roots of TKS during the developmental period that are involved in the pathways [14]. Proteome and transcriptome analysis showed that MVA is the key pathway for rubber synthesis, while the genes and proteins involved in the MEP pathway were expressed in low abundance. The rubber particle membrane is a continuous, monolayer membrane structure that provides a compatible interface between the rubber particles and the surrounding environment [15]. The small rubber particle protein (SRPP) and rubber elongation factor (REF) are the most important latex proteins in the rubber particle membrane [7]. SRPP/REF, Nogo-B receptor, and HRT1-REF bridging protein (HRBP), three rubber particles (RP) are membrane-binding proteins, are involved in the final elongation process of rubber synthesis in the TKS root [9,16]. SRPP covers the membrane surface and has a lower affinity for the membrane than REF, which is embedded in the membrane and attached to the rubber particles [17]. Both SRPP and REF interact with rubber particles and jointly regulate NRB. SRPP promotes polyisoprene synthesis, affects the molecular weight of rubber, and alters the length of rubber chains [18,19]. SRPP also plays a role in stabilizing rubber particles during rubber biosynthesis and can act as a latex coagulation factor in latex coagulation [20,21]. REF and SRPP proteins are highly homologous, and both share a common REF-conserved domain [10,22]. Compared to SRPP, the REF protein is smaller and has a partial sequence missing at the C-terminus [17]. It has been demonstrated that the overexpression of the *REF* gene is positively correlated with latex yield [23], and RNAi experiments on the REF gene resulted in a significant reduction in rubber yield, but did not affect the molecular weight of natural rubber, and the silencing of *REF* does not affect the regulation of rubber particle stability by *SRPP* [18].

Ethylene is an important phytohormone involved in the regulation of plant secondary metabolism, accelerating plant maturation, regulating photosynthesis and carbohydrate metabolism, and promoting chlorophyll degradation [24]. Ethylene is also involved in regulating the expression of genes related to biosynthesis such as flavonoid metabolism and anthocyanin glycosides [25]. Ethylene activates the laticifers regeneration pathway in *H. brasiliensis* [26] and inhibits the expression of rubber particle aggregase, which in turn affects latex yield [16,27,28]. Methyl jasmonate (MeJA), as well as other derivatives, can increase crop yields by stimulating secondary plant metabolism [29,30], and is involved in the latex regeneration pathway [7,31]. The *HbMADS4* gene of *H. brasiliensis* is induced by ethylene and MeJA significantly and the overexpression of *HbMADS4* in tobacco plants significantly represses the promoter activity of the *HbSRPP* gene [7]. The rubber tree is a woody plant; consequently, gene function studies have been slow to progress. Therefore, it is more appropriate to study the regulation mechanism of NRB in *T. kok-saghyz*, a model plant for rubber production. Because of the recent discovery that ethylene can effectively stimulate *H. brasiliensis* to enhance latex production, jasmonic acid (JA) and its volatile MeJA can be widely used as inducers of rubber laticifer differentiation [32]. However, the study of these two hormones on the same rubber-producing plant *T. kok-saghyz* has not yet been revealed and needs to be studied systematically and in-depth.

To systematically investigate the effect of *TkSRPP*/*REF* genes on NRB in *T. kok-saghyz*, in this paper, genome-wide identification of the *TkSRPP*/*REF* family members was carried out by using the available genome data. The physical and chemical properties, basic structure and evolutionary relationships of the family members were analyzed. In the qRT-PCR analysis of gene expression patterns in different tissues after ethylene and MeJA treatment, we observed the subcellular localization of part of the family genes. Furthermore, we carried out a preliminary identification of the main members of *TkSRPP*/*REF* involved in the regulation of natural rubber biosynthesis. These results provide new ideas for further research on exogenous phytohormones affecting natural rubber biosynthesis.

## 2. Results

### 2.1. Identification and Classification of SRPP/REF Gene Family in T. kok-saghyz

We obtained 8, 17, and 16 candidate *SRPP*/*REF* genes using three methods: keyword search, HMMER search, and BLASTP search, respectively. The motif, domain, repetitive sequences and transcripts of the same genes were screened for de-duplication. After primary screening and secondary identification, we finally identified a total of 12 *SRPP*/*REF* family members in the TKS genome. It contained 10 *SRPP* family members and 2 *REF* family members (Table 1). The results of the *SRPP*/*REF* sequence analyses showed that all the family genes contain typical REF structural domains (Pfam: PF05755), which belong to the REF family. Based on the *SRPP*/*REF* gene localization order of the TKS chromosome, they were named *TkSRPP1-TkSRPP10* and *TkREF1-TkREF2*. Subsequently, the physicochemical properties of each member of the *TkSRPP*/*REF* genes were analyzed (Table 1), such as the length of the open reading frame (ORF) of the gene sequences, molecular weight, amino acid number, and isoelectric point. The full lengths of the open reading frames of *TkSRPP*/*REF* were 381 bp (*TkSRPP1*) to 2121 bp (*TkREF2*); the number of amino acids was between 126 (*TkSRPP1*) and 706 (*TkREF2*). The results showed that *TkREF2* had the longest protein sequence with a molecular weight of 75.809 kDa, while *TkSRPP1* had the smallest molecular weight of 14.61 kDa. In addition, the isoelectric points (pl) of the *TkSRPP*/*REF* members ranged from 4.47 (*TkSRPP1*) to 8.79 (*TkREF2*). *TkSRPP1*, *TkSRPP7*, *TkSRPP8*, and *TkREF2* belong to the acidic proteins (pI < 7), and the rest are weak alkalinity proteins (pI > 7). Four TkSRPP/REF proteins had instability index values less than 40.0. The prediction of the aliphatic index of the family genes showed that TkSRPP/REF proteins have high stability, with *TkREF2* having the highest aliphatic index of 102.54 and TkSRPP3 having the lowest aliphatic index of 69.30. The grand average of hydropathicity (GRAVY) of the *TkSRPP*/*REF* family members was less than 0 except for *TkREF1*, which encodes hydrophilic proteins. The subcellular localization of the *TkSRPP*/*REF* genes family was predicted in the WoLF PSORT website. The results showed that only *TkREF1* was localized to the vesicular membrane and all the remaining members were localized in the cytoplasm (Table 1).

We predicted and analyzed the structure of the TkSRPP/REF proteins; the secondary structure of each member contained α-helix, β-turn, and random coil (Appendix A). Meanwhile, we predicted the three-dimensional structures of 12 TkSRPP/REF proteins using homology method modeling (Appendix A). All 12 proteins were detected and evaluated using online software. The secondary structure was dominated by α-helices, which accounted for 56.35–76.04%, followed by random coil accounting for 14.67–32.02%, whereas β-turn had the lowest percentage of 2.40–8.73% (Appendix A). A variety of secondary structures are grouped together to act as the basic building blocks of a tertiary structure (Appendix A). The large differences in the basic data of the *SRPP* subfamily and the *REF* subfamily suggest that members of the two subfamilies have experienced different evolutionary selective pressures, resulting in the exercise of slightly different biological functions.

### 2.2. Phylogenetic Analysis of TkSRPP/REF Proteins

To analyze the evolutionary relationship between *SRPP*/*REF* genes in different species, representative species were selected to construct a phylogenetic tree together with TKS. The *T. kok-saghyz*, *T. mongolicum*, *L. sativa*, *H. annuus,* and *G. max SRPP/REF* sequences were selected to align using Clustal W software, version 1.81, and a phylogenetic tree based on 1000 bootstrap replicates was constructed using the neighbor-joining method (Figure 1). According to the sequence similarity and phylogenetic analyses, the *SRPP*/*REF* genes were mainly divided into five evolutionary branches, with 13, 10, 16, 7, and 6 genes contained in evolutionary branches I to V, respectively. *TkSRPP7* and *TkSRPP8* were clustered in branch I (Figure 1). *TkSRPP4*, *TkSRPP5*, *TkSRPP9*, and *TkSRPP10* were clustered in branch II. *TkSRPP1*, *TkSRPP2*, and *TkSRPP3* were clustered in branch III. *TkREF1* and *TkREF2* were clustered in branch V. There were no family members presented in branch IV. In each branch, TkSRPP/REF first clustered with the orthologous species *T. mongolicum* and then with *L. sativa* and *H. annuus*. The SRPP/REF proteins of *G. max* were separated from the orthologous of other Asteraceae species. In addition, we further screened 111 SRPP/REF protein sequences from 17 species to construct a phylogenetic tree, and these genes were mainly divided into seven evolutionary branches. Branches IV and VI were not *TkSRPP*/*REF* genes, and *TkSRPP*/*REF* clustered more compactly with members of the Asteraceae family. This phenomenon suggests that *TkSRPP*/*REF* is distantly related to maize, rice, *Arabidopsis*, and other non-Asteraceae species. Furthermore, *TkREFs* were clustered in branch VII with other *REF* genes (Appendix A). We analyzed the motifs and conserved the structural domains and gene structures of the *TkSRPP*/*REF* genes. It was found that the basic structures of family members with close evolutionary relationships are relatively similar (Appendix A).

### 2.3. Chromosomal Localization of TkSRPP/REF Genes

To better show the evolutionary pattern of the *SRPP*/*REF* gene family, the genomic distribution of the *SRPP*/*REF* family genes in *T. kok-saghyz* and its related species *T. mongolicum* and *L. sativa* were mapped to chromosomal locations using TBtools software, version v2.096. According to genome data, *T. kok-saghyz* is diploid, and *T. mongolicum* and *L. sativa* are triploid. *T. kok-saghyz* has 16 chromosomes, and *T. mongolicum* and *L. sativa* have 24 chromosomes. The sequence information is all anchored to eight pseudochromosomes. The *SRPP* and *REF* genes have similar localization distributions, and the members of the *SRPP* and *REF* subfamilies are localized on different chromosomes, respectively (Figure 2).

All the *TkSRPP* subfamily members are localized on chromosome 4 in the *T. kok-saghyz* (Figure 2A), and two *TkREF* genes are localized on chromosome 5. Among them, there were nine *TkSRPP* members located in the gene cluster at the end of chromosome 4, centered between 1,204,419,836 bp and 123,368,525 bp, except *TkSRPP1*. These genes were closely aligned in position and functionally synergized with each other to regulate TKS secondary metabolic processes, whereas *TkSRPP1*, which was located far away from the gene cluster, is located at the other edge of chromosome 4 and positioned between 1,734,758 bp and 1,735,187 bp. *TkSRPP1* is far away from other *SRPP* subfamily members; it is suggested that this gene may possess an independent evolutionary branching in the course of evolution.

Seven *TmSRPPs* were localized on chromosome 4 (Figure 2B) and one *TmREF* was localized on chromosome 5. Similarly, *TmSRPP1* was localized between 1,018,788 bp and 1,020,349 bp at the end of chromosome 4 away from the gene cluster. *TmSRPP2*–*TmSRPP7* genes were localized in the gene cluster at the other end of chromosome 4, clustered between 72,592,019 bp and 72,760,006 bp. In *L. sativa*, *LaREF1* and *LaREF2* were localized on chromosome 8. *LaSRPPs* were localized on chromosome 9. Among them, *LaSRPP1* was localized between 5,548,999 bp and 5,550,612 bp at the end of chromosome 9, and the rest of the *LaSRPP* genes were clustered between 136,106,466 bp and 137,424,082 bp (Figure 2).

### 2.4. Gene Duplication, and Collinearity Analysis of TkSRPP/REF Genes

The *SRPP*/*REF* family was analyzed for collinearity and shown on chromosome loops (Figure 3). A total of three collinear blocks, including *TkSRPP2*/*TkSRPP9*, *TkSRPP3*/*TkSRPP10*, and *TkSRPP6*/*TkSRPP9*, were identified. The ratio of Ka/Ks between non-synonymous substitutions (Ka) and synonymous substitutions (Ks) for these three gene pairs was much lower than 1. This means that members of this family underwent a strong purifying selection (Table 2). Furthermore, we predicted replication events for *TkSRPP*/*REF* family members. The replication modes are more diverse. Only *TkSRPP1* had dispersed distribution, *TkSRPP8* had segmental duplication and the rest of the *TkSRPP* subfamily members underwent tandem duplication(Appendix A). However, both members of the *TkREF* subfamily are singleton genes. The results of replication event analysis indicate that *SRPP*/*REF* family members mainly rely on tandem duplication to expand members, and this mode of duplication is significant in the evolutionary process of the family. In addition, we have studied the collinearity analysis of *SRPP*/*REF* genes of TKS with other species. The results showed that there were two, three, five, two, and two homologous gene pairs of between *T. kok-saghyz* and *T. mongolicum*, *L. sativa*, *H. annuus*, *A. thaliana*, and *G. max*, respectively (Figure 4). Since *H. annuus* has a more complex genome, TKS has more homologous gene pairs with it.

### 2.5. Promoter Analysis of TkSRPP/REF Genes

Promoter cis-acting elements are important transcription initiation binding regions and play an important role in the regulation of gene expression. We used a 2.0 kb sequence upstream of the *TkSRPP*/*REF* gene promoter to predict cis-acting regulatory elements via the Plant CARE website (Figure 5).

Multiple classes of cis-acting elements were identified in the promoter regions of 12 *TkSRPP*/*REF* genes. These elements were classified into five major categories: promoter-related and binding sites, light-responsive, hormone-responsive, environment-responsive, and development-related. Among them, promoter-related and binding sites accounted for the largest proportion, ranging from 54.17% (*TkSRPP3*) to 77.95% (*TkSRPP10*) of all elements; environment-responsive elements ranged from 2.08% (*TkSRPP4*) to 11.46% (*TkSRPP3*); development-related elements ranged from 6.98% (*TkSRPP1*) to 15.63% (*TkSRPP3*); light-responsive was also present in all *SRPP*/*REF* promoter regions. *TkSRPP6* had the highest number of light-responsive elements, which accounted for 13.04% of all elements in this gene. The rest of the genes had light-responsive elements ranging from 2.11% (*TkSRPP9*) to 8.80% (*TkSRPP2*) (Figure 5A,B). In addition, hormone-responsive elements also accounted for a large proportion (Figure 5C), ranging from 3.47% (*TkSRPP4*) to 14.79% (*TkREF1*). All family genes contained Abscisic Acid (ABA)-response elements and most genes contained gibberellins (GA3)-, auxin (IAA)-, ethylene (ETH)-, methyl jasmonate (MeJA)- and Salicylic acid (SA)-response elements. The promoter regions of *TkSRPP*/*REF* family genes have MYB transcription factor binding sites related to drought inducibility, flavonoid biosynthesis gene regulation, and light response. These results suggest that members of the *TkSRPP*/*REF* family are likely to be involved in plant growth and metabolic regulatory processes.

### 2.6. Analysis of TkSRPP/REF Expression Patterns and Physiological Indexes

To further predict the function of *TkSRPP*/*REF* genes, we treated TKS with ethylene and detected the expression in different tissues using Real-Time Quantitative PCR. The results showed that the expression of *TkSRPP4*, *TkSRPP6*, *TkSRPP7*, *TkSRPP8* and *TkSRPP9* was significantly up-regulated after ethylene was induced in the roots. Among them, *TkSRPP6* and *TkSRPP8* were the most significantly up-regulated compared with the control, which is presumed to play a key role in the ethylene-regulated pathway. The addition of exogenous hormones activated the expression of these genes. *TkSRPP3*, *TkSRPP5*, *TkREF1*, and *TkREF2* were gradually down-regulated; *TkSRPP1* and *TkSRPP2* were significantly down-regulated in the 3 h of treatment, and *TkSRPP1* increased with the increase in treatment time. Meanwhile, the expression of *TkSRPP2* was significantly up-regulated at 6 h and then maintained at a lower level. Interestingly, *TkSRPP10* was up-regulated at the 3 h and then decreased, and up-regulated at 24 h again. In the leaves (Appendix A), the expression levels of most genes were increased, such as *TkSRPP1* and *TkSRPP5*, which were highly up-regulated in contrast to those in roots. The results of these experiments suggest that these genes are involved in the regulatory pathway of ethylene metabolism and regulate growth and developmental processes in specific tissues.

To understand the effect of ethylene treatment on the change in the rubber content of *T. kok-saghyz*, we determined the rubber content of plant roots at different times under treatment. As shown (Figure 6B), the yield of crude rubber extract showed a trend of a slight decrease followed by an increase. The yield peaked at 9.13% at the 24th hour. To evaluate the extent of cellular damage and the activation of the antioxidant defense system in *T. kok-saghyz* after ethylene treatment, we tested malondialdehyde (MDA) content as well as three key antioxidant enzyme activities: Peroxidase (POD), Catalase (CAT) and Superoxide Dismutase (SOD). TKS treated with ethylene showed an overall increasing trend in CAT activity compared to the control. However, the enzyme activity decreased at 6 h and then gradually increased to a higher level (Figure 6C). POD activity elevated significantly with treatment time (Figure 6D). At 6 h, SOD activities reached the highest level (Figure 6F).

In addition, we treated *T. kok-saghy* roots with MeJA and examined the expression of *TkSRPP*/*REF* genes (Figure 7 and Appendix A). It showed that most of the members were significantly up-regulated through MeJA induction. In roots, the level of up-regulation of these family members was very high, such as *TkSRPP4*, *TkSRPP7*, and *TkSRPP10* (Figure 7). The expressions of *TkSRPP2*, *TkSRPP6*, and *TkSRPP7* gradually increased with treatment time. Some genes had the highest expression level at 3 h. Similarly, in the leaves, the expression of some genes reached the highest level at 3 h (Appendix A). The expression of different members varies in different tissues. Only the expression level of *TkSRPP3* was reduced in leaves and maintained at a low level (Appendix A). The results indicated that this gene was significantly induced by MeJA and highly expressed in TKS roots. These experimental results suggest that the *TkSRPP*/*REF* family is involved in regulating the JA metabolic regulatory pathway.

In order to understand the effect of MeJA on the changes in *T. kok-saghyz* rubber content, we measured the crude rubber extract content of TKS roots under different treatment times. The results showed that the peak value was reached at 3 h after treatment, which was about 1.3 fold of the control (Figure 7B). After MeJA application, the CAT activity of TKS increased significantly in comparison to the control and peaked at 6 h (Figure 7C). POD activity reached three-fold of the control at 24 h (Figure 7D). MDA activity decreased slightly (Figure 7E). SOD activity first rose and then maintained within a range after a gradual decline (Figure 7F).

### 2.7. Subcellular Localization of TkSRPP/REF

To detect the location of *TkSRPP*/*REF* proteins in cells, we constructed 35S::eGFP, 35S::TkSRPP1-eGFP, 35S::TkSRPP4-eGFP, 35S::TkSRPP7-eGFP and 35S::TkREF1-eGFP fusion expression vectors and transiently expressed them in Nicotiana benthamiana (Figure 8). The subcellular localization of the *TkSRPP*/*REF* genes family was predicted in the WoLF PSORT website. The results showed that only TkREF1 was localized to the vesicular membrane, and all the remaining members were localized in the cytoplasm. *N. benthamiana* cells were observed using laser confocal microscopy and the fusion proteins all fluoresced green in the cytoplasm. We used mCherry red fluorescence as a cytoplasmic marker. As shown, 35S::TkSRPP1-eGFP, 35S::TkSRPP4-eGFP, 35S::TkSRPP7-eGFP and 35S::TkREF1-eGFP were expressed not only in the cytoplasm, but also in chloroplasts as well as the cytoplasmic membrane. It is hypothesized that this result is related to the function exercised by SRPP/REF proteins in the cytoplasm.

## 3. Discussion

Natural rubber has unique physicochemical properties and is widely used in industrial production, healthcare, civil engineering, construction, and other production research due to its strong abrasion resistance and good ductility [5,33]. *T*. *kok-saghyz* roots can produce natural rubber (NR), and the small rubber particle protein (SRPP) [34] and rubber elongation factor (REF) [35] are two important regulators in the mechanism of natural rubber production [17]. In this study, to deeply investigate the function of the *SRPP*/*REF* family in TKS, we conducted a genome-wide characterization of the *TkSRPP*/*REF* family based on information from the open data, and identified a total of 12 members of the *TkSRPP*/*REF* gene family as well as 16 other species (*T. mongolicum*, *H*. *brasiliensis*, *L. sativa*, *H. annuus*, *G. max*, *T. brevicorniculatum*, *C. annuum*, *S*. *italica*, *S. lycopersicum*, *C. scolymus*, *O. sativa*, *Z. mays*, *A. thaliana*, *P. vulgaris*, *M. truncatula* and *R. communis*) of 111 SRPP/REF proteins (Appendix A). The *SRPP*/*REF* family was divided into seven branches based on phylogenetic relationships (Appendix A). In different species, its members are unequally distributed in subclades. In particular, most of the family members of TKS are concentrated in subclades I and II with the closest affinity to the homologous species, *T. mongolicum. H. brasiliensis* are perennial woody plants. So, most of these *HbSRPP/REFs* are independently comprised in the subclade V. It has been shown that *TkSRPP* and *HbSRPP* belong to different evolutionary directions and play different roles in the rubber synthesis mechanism. In this study, we performed phylogenetic analyses for *SRPP*/*REF* family members in multiple species. Most of the rubber tree family members are independently located in subclade V and are genetically distant from TKS and other species, which is consistent with previous findings [36,37].

Variation in gene structure is an important feature of family evolution, and the analysis of the basic structure of *TkSRPP*/*REF* revealed that almost all family members have a three-exon, two-intron structure, except for *TkSRPP1*, which has a two-exon, one-intron structure, and *TkSRPP5*, which has a five-exon, four-intron structure (Appendix A). Due to the significant individual differences in the exon/intron structures of the family members, this difference enriches the gene functions of the *TkSRPP*/*REF* family [38]. TkSRPP/REF proteins contain a common REF-conserved structural domain at the N-terminus [34]. *TkREF1* has a special structure containing two incomplete REF-conserved structural domains, and it is hypothesized that TkREF1 plays a key role in the regulation of the NR synthesis [18]. The distribution of *SRPP*/*REF* family genes in *T. kok-saghyz* and its related species *T. mongolicum* and *L. sativa* were analyzed: the *SRPP*/*REF* genes of the three species had the same trend of differentiation. The *SRPP* subfamily genes and *REF* subfamily genes were located on different chromosomes, in which most of the genes in the *SRPP* subfamily existed in gene clusters, and only *SRPP1* was located far away from the gene clusters. The density of genes in the genome at the location of *SRPP1* is greater than that at the cluster. So, it is assumed that *SRPP1* plays an important role in the anabolism process of natural rubber.

We found that among the *TkSRPP* subfamily, only *TkSRPP1* has a deletion of its structural domain. This gene is located on the other end of chromosome 4 of TKS from other members of the *SRPP* subfamily (Figure 2), which is a dispersed distribution gene (Table 2) with a separate evolutionary branch. The expression of *TkSRPP1* is high in several tissues and the expression pattern is more diverse; these results are supported by previous studies [18,39,40] Due to the evolutionary specificity of the *TkSRPP1* gene, it is inferred that it has an important regulatory role in NR synthesis or metabolism. The remaining *TkSRPPs* genes are located in the gene cluster of chromosome IV; segmental duplication as well as tandem duplication modality play an important role in increasing the number of genes [41,42]. Most of the *TkSRPP*/*REF* family members are tandem duplicated, as this is the main mode of *TkSRPP*/*REF* family expansion. Meanwhile, the collinearity analysis of the family members calculated a total of three homologous gene pairs, each with Ka/Ks values less than 1. These genes have undergone a strong purifying selection and evolved in a more conserved manner [42].

Promoter region cis-acting elements are important regulators of plant growth, development, and resistance to biotic and abiotic stresses [43]. In order to further clarify the functions of the genes, we have analyzed the promoter regions of the *TkSRPP*/*REF* family. The cis-acting elements in the promoter region include various types, among which there are light-responsive elements (G-Box, MRE, etc.), hormone-responsive elements (GARE-motif, ABRE, ERE, etc.), growth and developmental elements (GCN4-motif, O2-site, etc.), defense and stress-responsive elements (STRE, WUN-motif, etc.), and abiotic stress elements (MYB, DRE, MBS, etc.). The *TkSRPP*/*REF* promoter region contains different kinds of hormone-response elements, such as ABA, IAA, JA, SA, etc., suggesting that they are likely to play roles in various hormone signaling pathways. In previous reports, *SRPP* gene expression was up-regulated by drought stress through the transcriptome analysis of Parthenium hysterophorus rubber particles (PR) [44]. *HbSRPP* was responsive to abiotic stresses such as hormones, cold, and heat [35]. *TbSRPP1* was an ABA-sensitive isoform involved in ABA signaling in response to drought, cold, and other abiotic stresses [45]. Ethylene is involved in inducing an increase in the phenolic acid content of *Salvia miltiorrhiza* hairy root [46]. Ethylene decreased the lignin content and increased the amount of secondary metabolites such as flavonoids in ramie [47]. MeJA activates the expression of Pyrethrin biosynthetic genes by inducing *TcMYC2* gene expression. The long-term induction of immature leaves with MeJA increased the accumulation of pyrethrin content [48]. Differentially expressed genes and the functional grading of *Catharanthus roseus* indicated that both MeJA and ethylene may stimulate the expression of genes related to vinblastine and vincristine production [49]. Since the gene function of *SRPP*/*REF* in TKS is not fully revealed, the prediction of cis-acting elements suggests that this family of genes not only plays a role in the rubber production mechanism, but most of them may respond to various biotic and abiotic stresses.

We used ETH as well as MeJA to treat *T*. *kok-saghyz* separately, and the *TkSRPP*/*REF* genes were differently expressed in different tissues. For example, the expression of *TkSRPP5* in roots was significantly reduced by ETH regulation, whereas the expression of this gene was higher in leaves under the same treatment. In roots (Figure 6), *TkSRPP6* and *TkSRPP8* were up-regulated by ethylene to a higher extent than in leaves at about 10-fold levels (Appendix A), and it is hypothesized that the reason for their ethylene-induced high expression in roots is to increase latex production [28]. Interestingly, the *TkSRPP4* and *TkSRPP10* genes do not contain ETH regulatory elements. However, both genes tended to be up-regulated in different tissues under ETH treatment. It is hypothesized that exogenous ETH application affected the transcript levels of *SRPP* upstream genes. In addition, the expression of almost all the genes of this family was significantly up-regulated at different times in both the root and leaf material treated by methyl jasmonate. *TkSRPP2* was most highly expressed in leaves at 3 h and in roots at 24 h. Only the expression of *TkSRPP3* in leaves was down-regulated by the stimulation of MeJA. It is possible that this gene functions differently in different tissues and is involved in the regulation of a variety of other metabolic regulatory pathways. Whether ETH and MeJA contain some kind of collaborative ability to co-regulate the synthesis of natural rubber remains to be thoroughly investigated. Based on this, we have a more comprehensive understanding of the *TkSRPP*/*REF* gene family and found that it may not only be involved in NR synthesis, but also may respond to a variety of biotic and abiotic stresses, and the specific metabolic regulation mechanism remains to be investigated.

## 4. Materials and Methods

### 4.1. Plant Materials and Phytohormone

The SHZ variety of *Taraxacum koksaghyz* was used as test material in this study. Samples were sterilized in 1.5% NaClO solution for 6 min and then placed in sterilized solid medium containing half Murashige and Skoog (1/2 MS). After 4 days of vernalization at 4 °C, they were placed in a growth chamber with a photoperiod of 16 h, a temperature of 25 °C and a relative humidity of 30%. After 14 d of germination period, the plants were transplanted into plastic pots with peat soil/vermiculite/perlite = 3:1:1 in the greenhouse. They were grown under a 16/8 h diurnal photoperiod at 25 °C. Six-month-old *T. koksaghyz* seedlings were selected as treatment material. Selected materials of the same size and growth were continued in Hoagland’s solution for 14 days. The selected seedlings were transferred into Hoagland solution containing 1 mmol/L MeJA and 100 μmol/L ethylene for hormonal treatment of *T. koksaghyz* roots. Three biological replicates were set up for each treatment with 12 *T. koksaghyz* seedlings per replicate. After five treatment periods of 0 h, 3 h, 6 h, 12 h and 24 h, the root and leaf materials of each replicate were collected and immediately placed in liquid nitrogen, and then stored at −80 °C for subsequent experimental analyses.

The alkaline boiling method was utilized to determine the rubber content of *T. kok-saghyz*. In total, 0.5 g of dried roots of *T. kok-saghyz* was taken and placed in a tube. Root tissues were completely immersed in the alkaline solution by adding 1 mol/L NaOH and boiling in a water bath for one hour. After a short time to cool down, a mortar and pestle was used to press the tissue into thin slices. Then, distilled water was added to wash the impurities. Distilled water was washed three times. Following this, 1% HCl solution was added and crude rubber was completely immersed in the acid solution in a boiling water bath for 15 min. The final washing of crude rubber was carried out using 95% ethanol. Finally, the washed crude rubber was dried in an oven at 65 °C until a constant weight. Malondialdehyde content, Peroxidase activity, Catalase activity, and Superoxide Dismutase activity were measured using the Solarbio Physiological Indicator Assay Kit (Solarbio, Beijing, China), Catalase (CAT) Activity Assay Kit (Solarbio, Beijing, China), Peroxidase (POD) Activity Assay Kit (Solarbio, Beijing, China), Malondialdehyde (MDA) Content Assay Kit (Solarbio, Beijing, China), and Superoxide Dismutase (SOD) Activity Assay Kit (Solarbio, Beijing, China).

### 4.2. Identification and Classification of SRPP/REF Gene Family in TKS

To identify the *T. koksaghyz SRPP*/*REF* genes, we obtained details of the latest genome of TKS through the website (https://ngdc.cncb.ac.cn/gwh/, accessed on 12 June 2023). The AtSRPP proteins were used as query sequences to retrieve SRPP/REF protein sequences in TKS. The E-value threshold for the BLAST program was set at 1 × 10^−10^ to obtain the candidate dataset of TkSRPP/REF proteins. The Hidden Markov Model (HMM) of the conserved structural domain of REF (PF05755) was downloaded from the Pfam website (http://pfam.xfam.org/, accessed on 14 June 2023), with the following threshold: e-values < 10^−5^. All putative *SRPP*/*REF* members with conserved REF or REF superfamily structural domains were found in our protein data using the HMM-search module in TBtools software at the following threshold: e-value of 10^−5^. Subsequently, the NCBI CDD (https://www.ncbi.nlm.nih.gov/Structure/cdd/, accessed on 14 June 2023) [50], InterProScan (http://www.ebi.ac.uk/Tools/pfa/iprscan/, accessed on 14 June 2023) and SMART (https://smart.embl.de/smart/batch.pl, accessed on 14 June 2023) databases were further checked for the integrity of the REF or REF superfamily structural domains of candidate *SRPP*/*REF* genes, and redundant sequences that did not contain the complete domains were removed (Table 1). Finally, 12 protein sequences with REF or REF superfamily structural domains were selected if their *SRPP*/*REF* protein homology was found to be extreme on the TKS chromosome. Their molecular weights (MW) and isoelectric points (pI) were predicted using ExPaSy (https://www.expasy.org/, accessed on 20 June 2023) and WoLF PSORT (https://www.genscript.com/wolf-psort.html, accessed on 20 June 2023), respectively, as was the subcellular localization. The secondary structures of the TkSRPP/REF proteins were predicted and analyzed using SOPMA website (https://npsa-pbil.ibcp.fr/cgi-bin/npsa_automat.pl?page=npsa_sopma.html/, access on on 22 June 2023) (Appendix A). The 3D structure of the protein was predicted via homology modeling using SWISS-MODEL (https://swissmodel.expasy.org/, accessed on 22 June 2023) [51,52,53,54,55]. Structure visualization was performed using 3D protein structure visualization PyMOL software, version 2.5.4.

### 4.3. Phylogenetic Analysis TkSRPP/REF Proteins and Gene Structure, Conserved Domains and Motif Composition of TkSRPP/REF Genes

All protein sequences of different taxa of species such as *Arabidopsis thaliana*, *Oryza sativa*, *Glycine max*, *T. mongolicum*, *H. annuus*, and *L. sativa* were downloaded from TAIR (https://www.arabidopsis.org/, accessed on 27 June 2023), Ensembl database (http://plants.ensembl.org/index.html, accessed on 27 June 2023), and Genome Archive database, respectively (https://ngdc.cncb.ac.cn/gwh/, accessed on 27 June 2023) and the National Centre for Biotechnology Information (https://www.ncbi.nlm.nih.gov/, accessed on 27 June 2023). Then, the sequences of *SRPP*/*REF* proteins in *Arabidopsis thalian*, *Oryza sativa*, and other species were obtained according to the methods described above (Appendix A). To explore evolutionary relationships, multiple sequence comparisons with default parameters were performed between 12 *T. kok-saghyz*, 10 *T. mongolicum*, 7 *Glycine max*, 11 *H. annuu*, and 12 *L. sativa SRPP*/*REF* proteins using Clustal W [56], followed by construction of phylogenetic trees with the neighbor-joining (NJ) method using MEGA11 software, version 11.0.11 [57]. To assess the reliability of the phylogenetic tree, the bootstrap value was set to 1000 repetitions. The evolutionary tree was visualized and annotated using the online software iTOL (https://itol.embl.de/, accessed on 1 July 2023). The motif composition and distribution of *TkSRPP*/*REFs* were identified using MEME [58] to resolve their conserved motifs. The parameters were as follows: optimum motif width set to ≥6 and ≤50; number of motifs: 10.

### 4.4. Chromosomal Distribution and Duplication Analysis of TkSRPP/REF Genes

Chromosome distributions in the identified *SRPP*/*REF* genes of *T. kok-saghyz*, *T. mongolicum* and *L. sativa* were obtained and visualized using Gene location visualize of TBtools software. Gene duplication events with default parameters were analyzed using the Mcscanx [59] to analyze collinearity between *SRPP*/*REF* by intraspecies versus interspecies (*T. kok-saghyz*, *T. mongolicum*, and *L. sativa*). Non-synonymous (Ka) and synonymous (Ks) substitutions were calculated for each duplicate *TkSRPP*/*REF* gene using the Simple Ka/Ks Calculator(NG) of TBtools software.

### 4.5. Promoter Analysis of TkSRPP/REF Genes

A 2000-bp sequence upstream of the start codon (ATG) was intercepted from the TKS reference genome using TBtools. Then, cis-acting elements in the promoter region were predicted using the online PlantCARE (http://bioinformatics.psb.ugent.be/webtools/plantcare/html/, accessed on 15 July 2023) [60]. Visualization and analysis of cis-acting elements of TKS *SRPP*/*REF* genes was performed using HeatMap of TBtools.

### 4.6. Quantitative Real-Time PCR (qRT-PCR) Analysis

Quantitative Real-Time PCR (qRT-PCR) was performed using TRIzol reagent (Invitrogen, Carlsbad, CA, USA) to extract total RNA from the samples. cDNA was synthesized using a FastQuant First Strand cDNA Synthesis Kit (Tiangen, Beijing, China) according to the manufacturer’s protocol. qRT-PCR was performed using a LightCycler 480 Real-Time PCR System (Roche, Basel, Switzerland), SYBR^®^ Green Premix Pro Taq HS qPCR kit (Accurate Biotechnology, Hunan, China) and Roche LightCycler instrument. Each treatment had three biological replicates. RT-PCR primers were designed using prime5 software, version v5.00 (PREMIER Biosoft, San Francisco, CA, USA) (Appendix A). *TkACTIN* was used as an internal reference gene; the primers used for qRT-PCR are shown in Appendix A. Data were analyzed using the 2^−∆∆Ct^ calculation method. Statistical analysis was performed using SPSS 22.0 software (SPPS Inc., Chicago, IL, USA). Statistical differences between measurements at different times or with different treatments were analyzed using Duncan’s multiple range test. Differences were considered significant at a probability level of *p* < 0.05.

### 4.7. Subcellular Localization of TkSRPP/REFs

Nicotiana benthamiana seeds were obtained from the Key Laboratory of Xinjiang Phytomedicine Resource and Utilization of Ministry of Education in Shihezi University (Shihezi, China). To study the transient expression of *TkSRPP*/*REFs* in *N*. *benthamiana* leaves, the full-length CDS of *TkSRPP*/*REFs* has been PCR-amplified using primers containing Sma I and Spe I restriction endonucleases (Appendix A) and ligated into the vector pCAMBIA1300-eGFP cleaved by Sma I and Spe I to generate pCAMBIA1300-35s-eGFP. The constructed vector was transformed into *Agrobacterium rhizogenes* GV3101 and infiltrated into 4-week-old *Nicotiana benthamiana* leaves, and the fluorescence signals in the epidermis of *N*. *benthamiana* leaves were observed using a confocal microscope (Nikon, Tokyo, Japan) after 48 h of dark incubation. The experiments used pCAMBIA1300-35S-mCherry-NOS (Puint, Shaanxi, China) as a marker for the cytoplasm.

## 5. Conclusions

In this study, a comprehensive analysis of the *T. kok-saghyz SRPP*/*REF* gene family was carried out, and a total of 10 *TkSRPPs* and 2 *TkREFs* were identified and classified. The basic physicochemical features, three-dimensional structural models, phylogeny, conserved structural domains, gene structure, chromosomal location, and cis-acting elements of the *TkSRPP*/*REFs* were analyzed by using genomic and bioinformatics techniques. Phylogenetic and homology analyses of *SRPP*/*REF* genes in different plants were used to provide new perspectives on the evolution of the *SRPP*/*REF* gene family in TKS. Finally, the tissue expression patterns of *TkSRPP*/*REFs* genes were analyzed under ethylene and MeJA treatments. This study provides useful information for the study of the *SRPP*/*REF* gene family in *T. kok-saghyz*, and these findings will contribute to a better understanding of the biological functions and molecular mechanisms of the *SRPP*/*REFs* in rubber substitute plants, and provide a basis for functional studies and molecular breeding for genetic improvement in *T. kok-saghyz*.

## Figures and Tables

**Figure 1 ijms-25-06864-f001:**
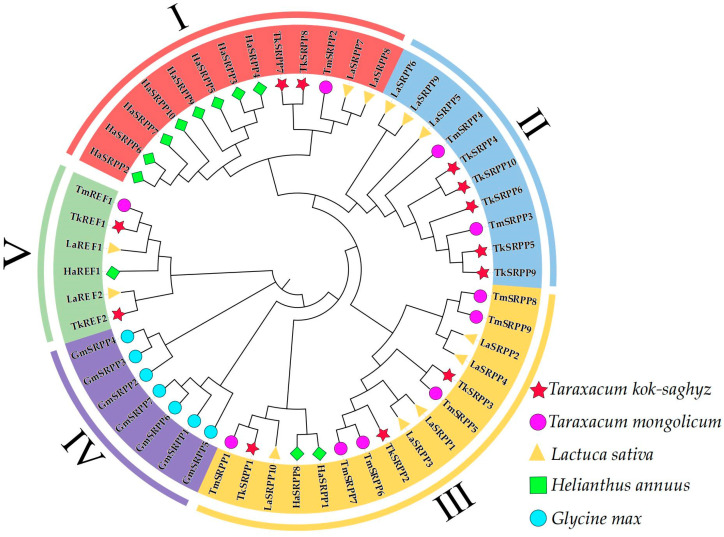
Phylogenetic analysis of TkSRPP/REFs proteins in *T. kok-saghyz*, *T. mongolicum*, *L. sativa*, *H. annuus* and *G. max*. Phylogenetic trees were plotted using the neighbor-joining (NJ) method with 1000 bootstrap replicates. Fifty-two genes were divided into five clades (I–V) and identified with different colors. The red pentagram represents *T. kok-saghyz*, the purple circle represents *T. mongolicum*, the yellow triangle represents *L. sativa*, the green square represents *H. annuus* and the blue circle represents *G. max*.

**Figure 2 ijms-25-06864-f002:**
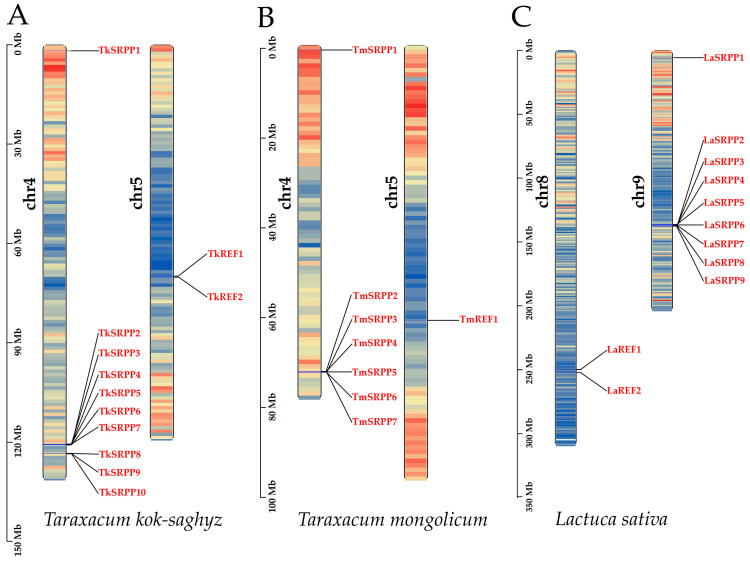
Chromosomal location of *SRPP*/*REF* genes in the *T. kok-saghyz* genome (**A**), and *T. mongolicum* genome (**B**) and *L. sativa* genome (**C**). The scale on the left represents the chromosome size, the length of chromosomes is measured in Mb. The chromosome number was indicated to the left of each chromosome. *TkSRPP*/*REF* gene numbers are shown on the right of each chromosome. The scale bar on the left indicates the genes of *TkSRPP*/*REF* are marked in red. Gene density of chromosomes from lower to higher is indicated from blue to red within the bar, respectively.

**Figure 3 ijms-25-06864-f003:**
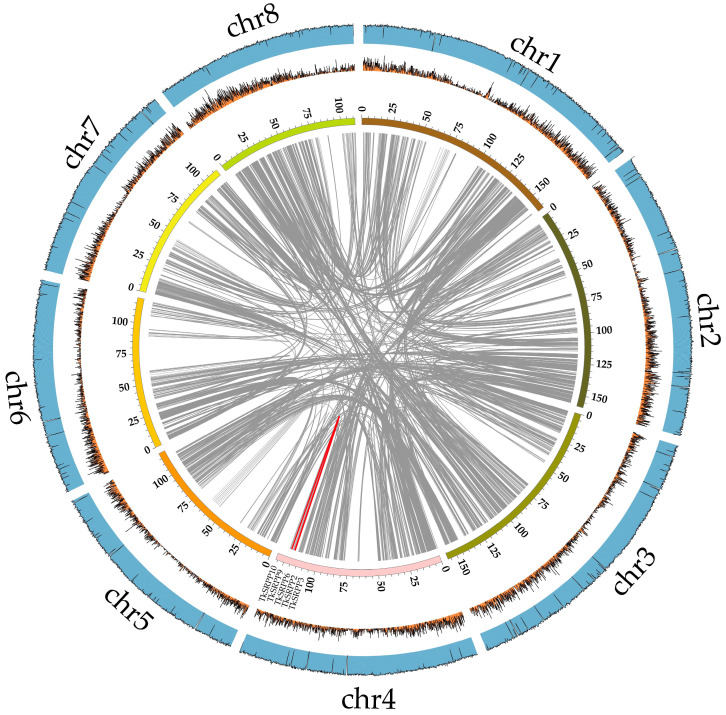
Collinearity analysis of *TkSRPP*/*REF* gene. The gray lines in the background represent colinear modules in the *T. kok-saghyz* genome. The red line indicates a *TkSRPP*/*REF* gene pair with collinearity. The outer bar chart represents gene density in chromosomes, the line chart represents GC content.

**Figure 4 ijms-25-06864-f004:**
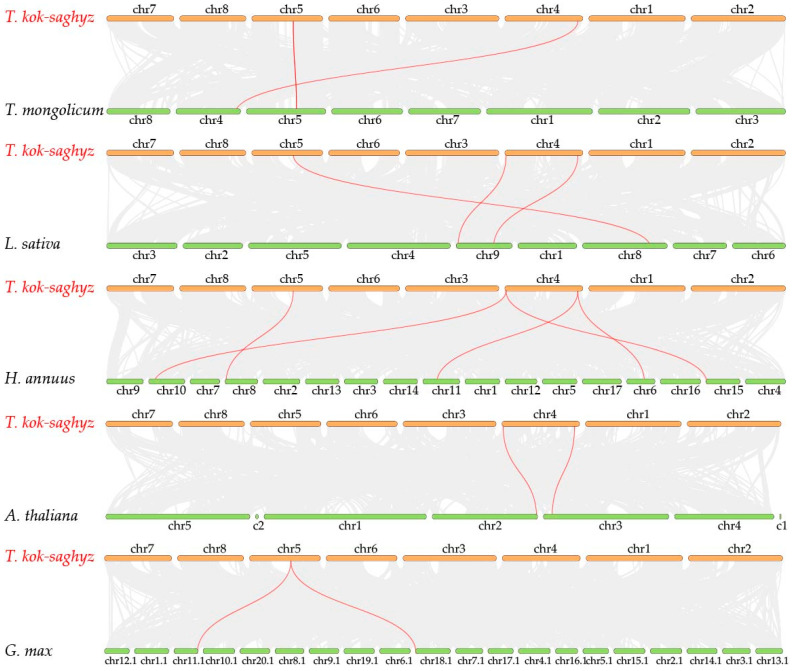
The colinear relationship between the *TkSRPP*/*REF* gene of *T. kok-saghyz* and the *SRPP*/*REF* genes of other species. The gray lines in the background represent colinear modules in the genomes of *T. kok-saghyz* and other plants. The highlighted red line indicates a colinear *SRPP*/*REF* gene pair.

**Figure 5 ijms-25-06864-f005:**
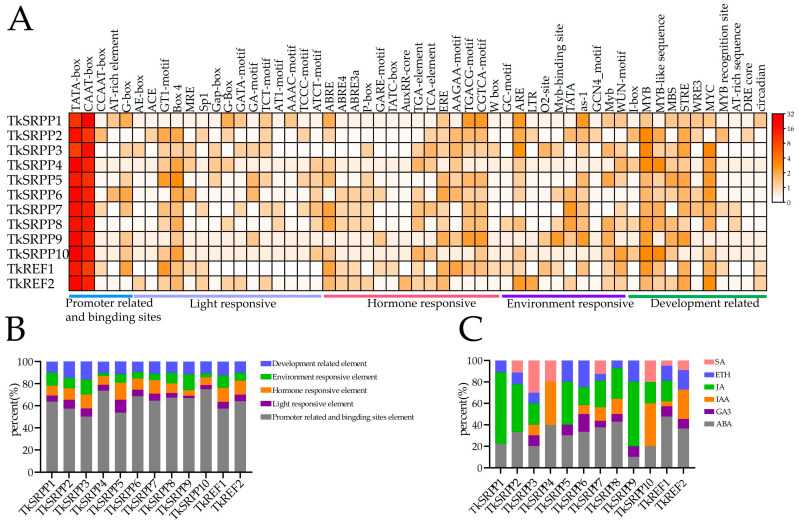
Analysis of cis-acting elements of *TkSRPP*/*REF* gene. (**A**) The different colors on the heatmap represent the number of different cis-acting elements in each *TkSRPP*/*REF* gene. (**B**) The different colors on the stacked graph represent the percentage of each type of cis-acting element in different *TkSRPP*/*REF* genes. (**C**) Different colors on the stacked graph represent the percentage of different hormone elements in different *TkSRPP*/*REF* genes.

**Figure 6 ijms-25-06864-f006:**
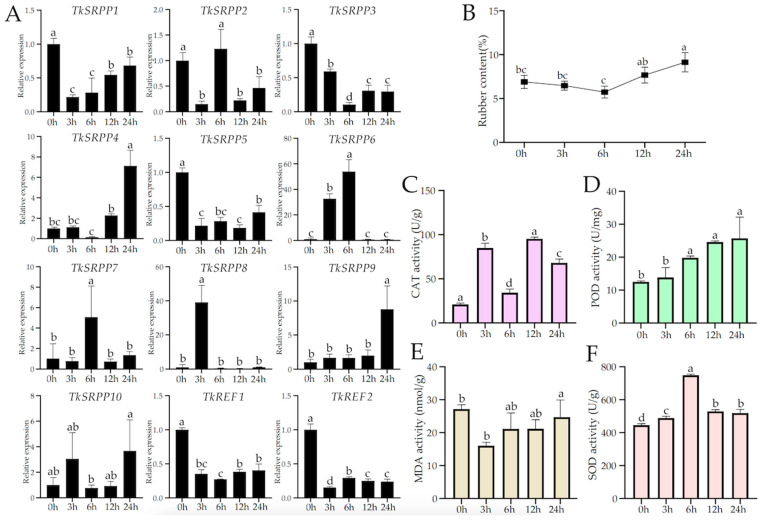
*TkSRPP*/*REF* expression pattern and physiological indexes under ethylene treatment. (**A**) Relative expression of *TkSRPP*/*REF* genes; (**B**) rubber content; (**C**) Catalase (CAT) activity; (**D**) Peroxidase (POD) activity; (**E**) Malondialdehyde (MDA) content; (**F**) Superoxide Dismutase (SOD) activity. The data are the average ± SD of three biological replicates. The error bar displays the mean ± SE of three independent replicates. The average values represented by the same letter showed no significant difference when *p* < 0.05, as determined using Duncan’s multiple range test.

**Figure 7 ijms-25-06864-f007:**
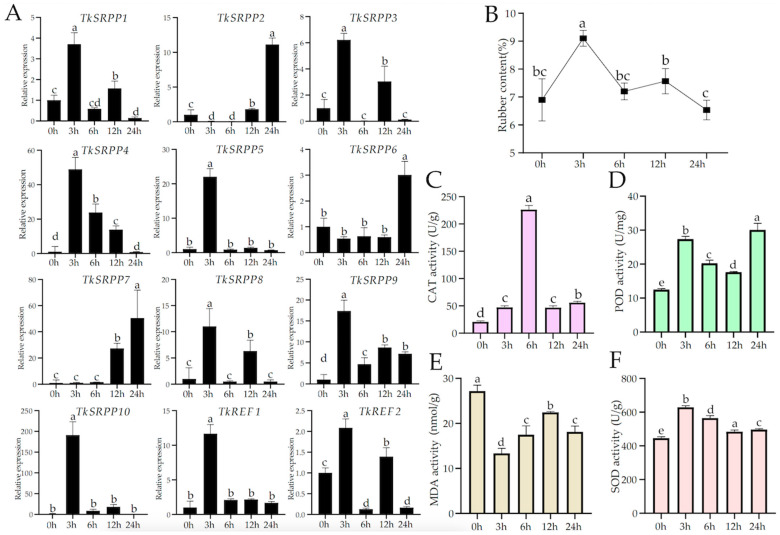
*TkSRPP*/*REF* expression pattern and physiological indexes under MeJA treatment. (**A**) Relative expression of *TkSRPP*/*REF* genes; (**B**) rubber content; (**C**) Catalase (CAT) activity; (**D**) Peroxidase (POD) activity; (**E**) Malondialdehyde (MDA) content; (**F**) Superoxide Dismutase (SOD) activity. The data are the average ± SD of three biological replicates. The error bar displays the mean ± SE of three independent replicates. The average values represented by the same letter showed no significant difference when *p* < 0.05, as determined using Duncan’s multiple range test.

**Figure 8 ijms-25-06864-f008:**
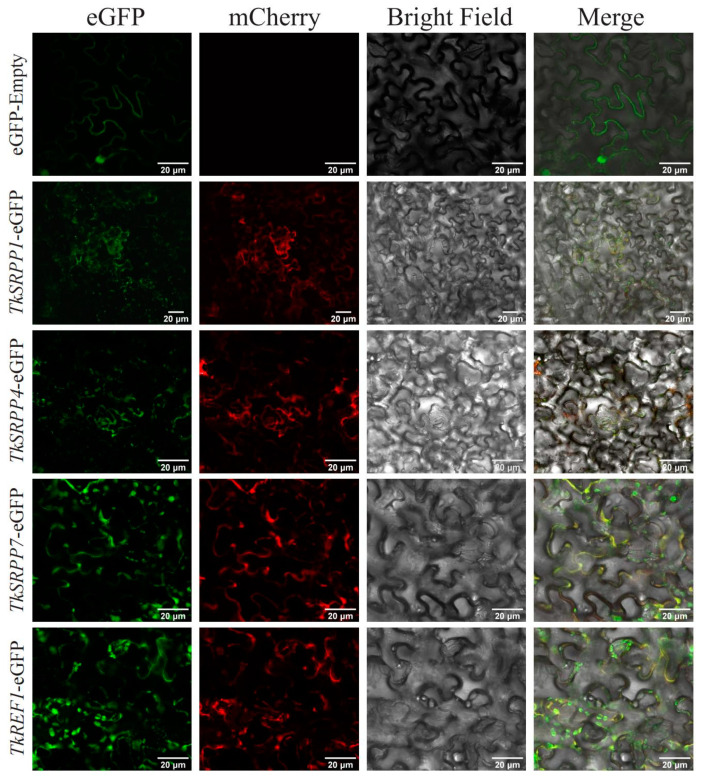
Subcellular localization. EGFP was the empty vector used as the control. All proteins were transiently expressed in *N*. *benthamiana* epidermal cells, and GFP signals were observed after 48 h of immersion. The cytoplasm was visualized with mCherry-labeled cytoplasm marker. Bars, 20 μm.

**Table 1 ijms-25-06864-t001:** Molecular characterization of *TkSRPP*/*REFs* in *T. kok-saghyz*.

	Gene ID	pI	MW (kDa)	aa	ORF/bp	Instability Index	Aliphatic Index	GRAVY	Subcellular
*TkSRPP1*	*GWHGBCHF021617*	8.54	14.61	126	381	43.84	79.76	−0.492	cyto
*TkSRPP2*	*GWHGBCHF026516*	4.47	25.166	235	708	38.78	87.87	−0.245	cyto
*TkSRPP3*	*GWHGBCHF026517*	4.75	24.547	228	687	40.76	69.30	−0.485	cyto
*TkSRPP4*	*GWHGBCHF026520*	5.35	25.351	232	699	47.80	76.51	−0.355	cyto
*TkSRPP5*	*GWHGBCHF026521*	5.34	25.829	236	711	52.70	78.52	−0.330	cyto
*TkSRPP6*	*GWHGBCHF026522*	5.68	25.284	232	699	43.56	78.19	−0.352	cyto
*TkSRPP7*	*GWHGBCHF026523*	8.79	23.174	208	627	39.04	89.90	−0.201	cyto
*TkSRPP8*	*GWHGBCHF026613*	8.50	23.169	208	627	41.82	89.42	−0.206	cyto
*TkSRPP9*	*GWHGBCHF026615*	5.44	25.286	232	699	50.44	77.76	−0.319	cyto
*TkSRPP10*	*GWHGBCHF026616*	5.35	25.351	232	699	47.80	76.51	−0.355	cyto
*TkREF1*	*GWHGBCHF028815*	4.92	44.632	409	1230	28.71	102.54	0.050	vacu
*TkREF2*	*GWHGBCHF028818*	8.61	75.809	706	2121	29.41	97.61	−0.006	cyto

pI, isoelectric points; MW, molecular weight; aa, amino acid number; ORF, open reading frame; GRAVY, grand average of hydropathicity; Cyto, cytoplasmic; Vacu, vacular membrane.

**Table 2 ijms-25-06864-t002:** Homologous gene pair Gene 1 and Gene 2, Ka, Ks and Ka/Ks of the *TkSRPP*/*REF* gene family.

Gene 1	Gene 2	Ka-Value	Ks-Value	Ka/Ks-Value
*TkSRPP3*	*TkSRPP10*	0.296318810632272	1.00829564953446	0.293880877864626
*TkSRPP6*	*TkSRPP9*	0.015665126442706	0.08216362065824	0.190657693967318
*TkSRPP2*	*TkSRPP9*	0.376302584848833	1.25186786035179	0.300592895437931

## Data Availability

All data generated or analyzed during this study are included in this article.

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
