# Peer review of "Genome-Wide Analysis of the SRPP/REF Gene Family in Taraxacum kok-saghyz Provides Insights into Its Expression Patterns in Response to Ethylene and Methyl Jasmonate Treatments"

_ijms, 2024, doi:10.3390/ijms25136864_

Round 1
Reviewer 1 Report
Comments and Suggestions for Authors
First of all, this manuscript is undoubtedly within the scope of the IJMS journal. After a detailed reading, my first impression is that the manuscript provides valuable new insights in the topic, but I have difficulties with the language. Likewise, the manuscript has technical and typo errors. It was unprofessional to send the manuscript to a Q1 journal in this form, with non-fluent English. For this reason, I suggest that the EiC request a major revision with necessary English proofreading and extra time to polish the technical issues in the manuscript. After that, I will read the entire manuscript and provide line-by-line suggestions. In its current form, this exceeds the standard revision process.
Comments on the Quality of English LanguageLanguage is not fluent as expected in Q1 journals.
Author Response
Thank you very much to the reviewers for pointing out the errors and suggestions. I apologize for any inconvenience caused. I have revised the manuscript. Please check it.For this manuscript, I have revised it word by word to ensure the fluency of English. I have also corrected some specialized vocabulary and proofread the abbreviation of words and the way of writing species. Thank the reviewers for their very professional review of this manuscript. Your professional review is of great help to improve the research level of this manuscript.
Reviewer 2 Report
Comments and Suggestions for Authors
The authors investigate the function of the small rubber particle protein (SRPP) and rubber elongation factor (REF) family in Taraxacum kok-saghyz and identified and classified a total of 12 members of the SRPP and REF gene family. Generally, the manuscript is good, the introductions is sufficient and informative. The methods are clearly described, and the discussion is deep enough. However, I have some concerns:
I recommend the manuscript to be considered for publication after minor revision.
L 106-108: The sentence must be revised.
L 106-108: The sentence does not fit into the introduction chapter. Here, as the last part of this chapter, the purpose of the study should be specified.
L 111-112: This phrase is not properly formulated, must be revised.
L 192-200: The sentence must be revised, it is difficult for the reader to follow.
Fig. 4, L 250: There is no highlighted blue line!
L 273-274: please specify the full name, not only the abbreviation, for the words that appear for the first time in the text.
Author Response
(1)L 106-108: The sentence must be revised.
Thank you very much for your helpful suggestion, this sentence has been revised in the reworked manuscript.
(2)L 106-108: The sentence does not fit into the introduction chapter. Here, as the last part of this chapter, the purpose of the study should be specified.
Thank you very much for the reminder that this sentence has been condensed to suit the essay. The details can be found in the returned manuscript. Thanks again for the heads up.
(3)L 111-112: This phrase is not properly formulated, must be revised.
Thank you very much for your kind advice, the problem with the presentation of this sentence has been rewritten in the return manuscript.
(4)L 192-200: The sentence must be revised, it is difficult for the reader to follow.
Many thanks to the reviewer's good advice, this sentence was rewritten in the reworked manuscript.
(5)Fig. 4, L 250: There is no highlighted blue line!
Thank you very much for the heads up, we have checked the color issue in the figure note and completed the revision in the return manuscript.
(6)L 273-274: please specify the full name, not only the abbreviation, for the words that appear for the first time in the text.
Thank you very much for the heads up, we have made corrections one by one according to your suggestions. The first occurrence of words in the original lines L273-274 have been replaced with the full name, ABA replaced with Abscisic Acid (ABA), GA3 replaced with Gibberellin A3 (GA3), IAA replaced with auxin (IAA), and SA replaced with Salicylic acid (SA). When our was revising, and found the same error in L270 and L272 above, so we also revised them together, JA was replaced with jasmonic acid(JA), and ABA was replaced with Abscisic Acid(ABA) in L272. Thanks again for your guidance.
Reviewer 3 Report
Comments and Suggestions for Authors
The authors performed a genome-wide analysis of the SRPP/REF gene family of Russian dandelion and identified a total of 10 TkSRPP genes and 2 TkREF genes, which might play crucial roles in natural rubber synthesis. They also analyzed the evolutionary relationships of SRPP/REF genes among different species, chromosomal localization, gene duplication, collinearity, and cis-acting elements at the promoter regions, etc. The aim of this study is important. Because a better understanding of the mechanism underlying the production of natural rubber affected by the expression of the rubber biosynthesis genes will guide the molecular breeding of TKS in the future.
However, there are too many weaknesses in this manuscript.
First, the manuscript is poorly written in terms of English language quality. The manuscript is filled with numerous easily noticeable errors. 1) incomplete sentences, such as “TkSRPP6 and TkSRPP8 were abundantly expressed in ethylene-induced, and ….” in the abstract, “The systematic analysis of classification, protein secondary structure, three-dimensional structure, phylogenetic analysis, gene structure, chromosomal localization, collinearity, cis-elements, and of this gene family.” at Page 3 Line 106-108, etc. 2) Incorrect punctuation, such as Page 2 Line 66, Line 89, etc. 3) Redundant sentence, such as Page 2 Line 74-75,” REF protein sequence is highly homologous to SRPP protein sequences. REF and SRPP proteins are highly homologous”. 4) elusive statement, such as “inhibits the expression of the enzyme rubber particle aggregate that enhance the yield of latex” at Page 2 Line 91, “Therefore, it is necessary to use TKS a model plant for rubber production, to study the regulation mechanism of NRB is more suitable.” at Page 3 line 97-98. etc. 5) Grammatical mistakes throughout the text, such as “at the 6-hour” instead of “in the 6h”, etc. Clearly, the authors didn’t review the manuscript carefully before the submission.
The author needs to read the original paper before they cite a paper which supports their statement. For example, they cited a paper entitled “HbMADS4, a MADS-box Transcription Factor from Hevea brasiliensis, Negatively Regulates HbSRPP.” In this paper, it is found that the HbMADS4 gene of Hevea brasiliensis is induced by ethylene and MeJA and the overexpression of HbMADS4 in tobacco plants significantly represses the promoter activity of the HbSRPP gene. However, the effect on NR synthesis was not determined in this paper.
The authors in this manuscript made a common mistake of overstating the conclusion drawn by cited papers. I suggest the authors check other references carefully.
The authors analyzed cis-acting elements at the promoter regions of these genes. In addition to JA and ETH-acting elements, there are other elements such as ABA, GA, IAA and SA. It would be a more comprehensive study if they could analyze the responsive expression of these genes with treatment of other hormones.
The authors analyzed the subcellular localization of the selected SRPP and REF proteins using confocal microscope. The information about this part is incomplete and no results were described, such as subcellular localization, the correlation of the localization and their potential function. The methods and materials used for this assay are also incomplete, such as mCherry control, tobacco plants, ect. No markers were used to characterize the subcellular localization.
Comments on the Quality of English Language
Must be improved.
Author Response
1) incomplete sentences, such as “TkSRPP6 and TkSRPP8 were abundantly expressed in ethylene-induced, and ….” in the abstract, “The systematic analysis of classification, protein secondary structure, three-dimensional structure, phylogenetic analysis, gene structure, chromosomal localization, collinearity, cis-elements, and of this gene family.” at Page 3 Line 106-108, etc.
Thank you very much for the reviewer's suggestion, we have already finished revising the problem in this sentence in the reworked manuscript.
2)Incorrect punctuation, such as Page 2 Line 66, Line 89, etc.
Thank you very much for your kind reminder, the symbol issue in the article has been completed and revised in the return manuscript. Thank you again for your reminder.
3) Redundant sentence, such as Page 2 Line 74-75,” REF protein sequence is highly homologous to SRPP protein sequences. REF and SRPP proteins are highly homologous”.
Thank you very much for the helpful reminder that the deletion of redundant sentences has been completed. The details are reflected in the reworked manuscript.
4) elusive statement, such as “inhibits the expression of the enzyme rubber particle aggregate that enhance the yield of latex” at Page 2 Line 91, “Therefore, it is necessary to use TKS a model plant for rubber production, to study the regulation mechanism of NRB is more suitable.” at Page 3 line 97-98. etc.
Thank you very much for the reviewer's suggestion, we have completed the revision of this sentence and it is reflected in the return manuscript.
5) Grammatical mistakes throughout the text, such as “at the 6-hour” instead of “in the 6h”, etc. Clearly, the authors didn’t review the manuscript carefully before the submission.”
Thank you so much for your helpful advice, we found many errors in P3 and P6 according to your suggestion, and similar problems occurred in L293, L295, L296, L297, L298, L299, L230, L303, L319, L320 were modified one by one. Once , Thank you very much again for your busy schedule to put forward valuable suggestions.
6) The author needs to read the original paper before they cite a paper which supports their statement. For example, they cited a paper entitled “HbMADS4, a MADS-box Transcription Factor from Hevea brasiliensis, Negatively Regulates HbSRPP.” In this paper, it is found that the HbMADS4 gene of Hevea brasiliensis is induced by ethylene and MeJA and the overexpression of HbMADS4 in tobacco plants significantly represses the promoter activity of the HbSRPP gene. However, the effect on NR synthesis was not determined in this paper.
Thank you very much for your good advice, we used more appropriate words to describe the results of this section. Thanks again.
7) The authors in this manuscript made a common mistake of overstating the conclusion drawn by cited papers. I suggest the authors check other references carefully.
Many thanks to the reviewers for their kind reminders. The results of the cited sections and references have been checked during this article revision process. Thank you again for your helpful reminder.
8) The authors analyzed cis-acting elements at the promoter regions of these genes. In addition to JA and ETH-acting elements, there are other elements such as ABA, GA, IAA and SA. It would be a more comprehensive study if they could analyze the responsive expression of these genes with treatment of other hormones.
Many thanks to the reviewers for your suggestions, which are very correct. In the early stages of the experiment, we had thoughts similar to your suggestion. Because there are clear studies showing that ethylene promotes the biosynthesis of natural rubber in rubber trees, thus increasing the rubber content. Methyl jasmonate stimulates the activity of jasmonic acid-related transcription factors, thereby affecting the expression of genes in the natural rubber production mechanism. It is hypothesized that ethylene and methyl jasmonate can lead to the expression of genes in the natural rubber production mechanism in response to the hormone. Therefore, two phytohormones, ethylene and methyl jasmonate, were selected to treat T. kok-saghyz roots in this thesis. Other hormone responses will be continued in subsequent studies to investigate the function of SRPP/REF genes in addition to natural rubber production.
Reference 1: Liu J P , Zhuang Y F , Guo X L ,et al.Molecular mechanism of ethylene stimulation of latex yield in rubber tree (Hevea brasiliensis) revealed by de novo sequencing and transcriptome analysis[J].BMC Genomics, 2016.DOI:10.1186/s12864-016-2587-4.
In Ref. 1: Ethylene stimulates rapid synthesis of precursors for natural rubber biosynthesis in rubber trees. Increased rubber yield by ethylene treatment may be associated with up-regulation of key enzymes of the glycolytic pathway and C3 nitrogen fixation.
Reference 2: Cao, X.W., Yan, J., Lei, J., Li, J., Zhu, J. and Zhang, H., 2017. De novo Transcriptome Sequencing of MeJA-Induced Taraxacum koksaghyz Rodin to Identify Genes Related to Rubber Formation. Sci Rep 7, 15697.
In Ref. 2: Methyl jasmonate regulates rubber biosynthesis-related genes in TKS by affecting jasmonic acid-related transcription factors, which further regulate rubber secondary metabolism.
9)The authors analyzed the subcellular localization of the selected SRPP and REF proteins using confocal microscope. The information about this part is incomplete and no results were described, such as subcellular localization, the correlation of the localization and their potential function. The methods and materials used for this assay are also incomplete, such as mCherry control, tobacco plants, ect. No markers were used to characterize the subcellular localization.
Thank you very much for the reviewer's reminder. The description of the results of subcellular localization has been added accordingly based on your suggestion. At the same time, the material and method section has been improved. The details are reflected in the reworked manuscript. Thank the reviewers for their very professional review of this manuscript. Your professional review is of great help to improve the research level of this manuscript.
Reviewer 4 Report
Comments and Suggestions for Authors
The author conducted a study on the Genome-wide Characterization and Eth and MeJA Induced Expression Patterns of the SRPP/REF Gene Family in Taraxacum kok-saghyz. The paper is innovative and well-analyzed, but it has the following issues:
The title needs to be further revised to align with the content of the manuscript.
line 24. The second occurrence of the Latin name of a plant requires an abbreviation. etc.
Line 27: "up-regulated at different treatment times by MeJA-induced root and leaf." What does this mean? Is it “Expression of genes was up-regulated in roots and leaves at different time points following MeJA treatment?”
lines 52-53, using Latin abbreviations for plants, using abbreviations like "tks" "tb"can confuse the reader.
Line 90 Hevea brasiliensis abbreviation, etc.
Line 155: "The SRPP/REF protein sequences of T. kok-saghyz, T. mongolicum, L. sativa, H. annuus, and G. max were compared using Clustal W software." It should be "align" instead of "compare."
Line 174 Abbreviation for plant latin name“Arabidopsis thaliana and 174Hevea brasiliensis”.
How many chromosomes does T. kok-saghyz have? This should be reflected in the description.
Lines 212-213 should use italics for Latin.
Figure 3 should only reflect the collinearity of target genes within species.
Lines 277-278: "These results suggest that TkSRPP/REF is not only involved in the growth and development of TKS, but also plays a key role in biotic and abiotic stresses." Is this appropriate? Mainly involved in the synthesis of secondary metabolites, is there ref support for its relationship with the growth and development of TKS?
Figure 6: 0h does not mark error lines.
The discussion section should not merely repeat the results. It should be reorganized to offer deeper insight, especially as the author addresses Eth and JAs. There are many refs about Eth and JAs in regulating secondary metabolites, such as the MeJA-induced TcMYC2, TcWRKY75 regulates the synthesis of pyrethrins, and MeJA enhances the synthesis of artemisinin, etc.
Section 4.5 on plant materials should be introduced earlier.
How were the concentrations of Eth and MeJA determined?
529 Specific Latin name for tobacco.
538 There is a problem with the statistical analysis, t-test cannot be used for comparisons among multiple samples.
Author Response
The title needs to be further revised to align with the content of the manuscript.
Thank you very much for your helpful suggestion. We further revised the title of the manuscript according to the research content to make the title consistent with the research content. Thank you again for your reasonable suggestions.
(1)line 24. The second occurrence of the Latin name of a plant requires an abbreviation. etc.
Thank you for your kind reminder that the issue of abbreviated plant names has been revised and the specifics are reflected in the reworked manuscript.
(2) Line 27: "up-regulated at different treatment times by MeJA-induced root and leaf." What does this mean? Is it “Expression of genes was up-regulated in roots and leaves at different time points following MeJA treatment?”
Thank you very much for your helpful suggestion. This sentence has been rewritten, and the specifics are reflected in the reworked manuscript.
(3)lines 52-53, using Latin abbreviations for plants, using abbreviations like "tks" "tb"can confuse the reader.
Thank you for your kind reminder that the issue of abbreviated plant names has been revised and the specifics are reflected in the reworked manuscript.
(4)Line 90 Hevea brasiliensis abbreviation, etc.
Thank you very much for your helpful suggestion that the issue of abbreviated plant names has been revised and the specifics are reflected in the reworked manuscript.
(5)Line 155: "The SRPP/REF protein sequences of T. kok-saghyz, T. mongolicum, L. sativa, H. annuus, and G. max were compared using Clustal W software." It should be "align" instead of "compare."
Thank you to the reviewer for the heads up and we have finished revising this error. Thanks again for your reminder.
(6)Line 174 Abbreviation for plant latin name“Arabidopsis thaliana and 174Hevea brasiliensis”.
Thank you for your kind reminder that the issue of abbreviated plant names has been revised and the specifics are reflected in the reworked manuscript.
(7)How many chromosomes does T. kok-saghyz have? This should be reflected in the description.
Thank you for your kind reminder to add T. kok-saghyz chromosome number and ploidy to result 2.3 as per your suggestion. Also add the chromosome and ploidy of T. mongolicum and L. sativa together to result 2.3. Thank you again for your helpful suggestion.
(8)Lines 212-213 should use italics for Latin.
Thank you for your kind reminder that the issue of abbreviated plant names has been revised and the specifics are reflected in the reworked manuscript.
(9)Figure 3 should only reflect the collinearity of target genes within species.
Thank you very much for your good suggestion. But I think the co-linearity within the genome is part of what should be in this picture. We just highlighted the SRPP/REF genes in the T. kok-saghyz genome. The rest of the lines are to represent all gene pairs within the T. kok-saghyz reference genome. There are reservations about this part in similar gene family identification papers, for example:
- Yan C, Yang T, Wang B, Yang H, Wang J, Yu Q. Genome-Wide Identification of the WD40 Gene Family in Tomato (Solanum lycopersicum L.). Genes (Basel). 2023;14(6):1273. doi:10.3390/genes14061273
Figure 4A in this paper is consistent with Figure 3 in this paper.
- Shen Y, Liu Y, Liang M, Zhang X, Chen Z, Shen Y. Genome-Wide Identification and Characterization of the Phytochrome Gene Family in Peanut. Genes (Basel). 2023;14(7):1478. doi:10.3390/genes14071478
Figure 4 in this paper is consistent with Figure 3 in this paper.
3. Wang R, Zhao W, Yao W, Wang Y, Jiang T, Liu H. Genome-Wide Analysis of Strictosidine Synthase-like Gene Family Revealed Their Response to Biotic/Abiotic Stress in Poplar. Int J Mol Sci. 2023;24(12):10117. doi:10.3390/ijms241210117
Figure 3 in this paper is consistent with Figure 3 in this paper.
Thank you very much again for your kind suggestion.
(10)Lines 277-278: "These results suggest that TkSRPP/REF is not only involved in the growth and development of TKS, but also plays a key role in biotic and abiotic stresses." Is this appropriate? Mainly involved in the synthesis of secondary metabolites, is there ref support for its relationship with the growth and development of TKS?
Thank you so much for the reminder, we have carefully revised my manuscript based on your suggestion to put the paragraph in more appropriate words.
(11)Figure 6: 0h does not mark error lines.
Thank you very much for the reminder, we recalculated the error values for each set of data and generated the error bars. Corrections involving the same errors have been completed for Fig6, Fig7, FigS4 and FigS5.
(12)The discussion section should not merely repeat the results. It should be reorganized to offer deeper insight, especially as the author addresses Eth and JAs. There are many refs about Eth and JAs in regulating secondary metabolites, such as the MeJA-induced TcMYC2, TcWRKY75 regulates the synthesis of pyrethrins, and MeJA enhances the synthesis of artemisinin, etc.
Thank you very much for your kind reminder of this very important suggestion. After reviewing the relevant papers, we have added the basis for the induction of secondary metabolites by ethylene and methyl jasmonate to produce changes in plants or genes. The details are reflected in the reworked manuscript.
(13)Section 4.5 on plant materials should be introduced earlier.
Thank you very much for the reminder, the plant material in the methodology has been revised to section 4.1. The details are reflected in the reworked manuscript.
(14)How were the concentrations of Eth and MeJA determined?
Many thanks to the reviewer for the question. This issue has been a concern in the early stages of the thesis research. It is known from the literature that ethylene can stimulate the rapid synthesis of natural rubber synthesis precursors in rubber trees, thus enhancing rubber yield. Methyl jasmonate has been shown to regulate rubber secondary metabolism and affect rubber biosynthesis. Therefore, two plant hormones, ethylene and methyl jasmonate, were used to treat T. kok-saghyz roots in this study.
Reference 1: Liu J P , Zhuang Y F , Guo X L ,et al.Molecular mechanism of ethylene stimulation of latex yield in rubber tree (Hevea brasiliensis) revealed by de novo sequencing and transcriptome analysis[J].BMC Genomics, 2016.DOI:10.1186/s12864-016-2587-4.
In Ref. 1: Ethylene stimulates rapid synthesis of precursors for natural rubber biosynthesis in rubber trees. The ethylene treatment, which increased rubber yield, may be related to the up-regulation of key enzymes of the glycolytic pathway and C3 nitrogen fixation.
Reference 2: Cao, X.W., Yan, J., Lei, J., Li, J., Zhu, J. and Zhang, H., 2017. De novo Transcriptome Sequencing of MeJA-Induced Taraxacum koksaghyz Rodin to Identify Genes Related to Rubber Formation. Sci Rep 7, 15697.
In Ref. 2: This article uses MeJA solution sprayed on T. kok-saghyz leaves for treatment. The results indicate that methyl jasmonate affects rubber biosynthesis-related genes in T. kok-saghyz through the regulation of jasmonate-related transcription factors, which in turn affects rubber biosynthesis-related genes in T. kok-saghyz. Thus, it regulates rubber secondary metabolism.
Reference 3: G. Dong, H. Wang, J. Qi, Y. Leng, J. Huang, H. Zhang, J. Yan, Transcriptome analysis of Taraxacum kok-saghyz reveals the role of exogenous methyl jasmonate in regulating rubber biosynthesis anddrought tolerance, Gene Gene (2023), doi: https://doi.org/10.1016/j.gene.2023.147346
In Ref. 3: The data in this article was also treated using MeJA solution sprayed on T. kok-saghy leaves.
Thank you very much for your kind suggestion.
(15) 529 Specific Latin name for tobacco.
We appreciate your kind reminder that the plant name has been changed to the Latin name. The details are reflected in the reworked manuscript. Thank you again for your kind reminder.
(16)538 There is a problem with the statistical analysis, t-test cannot be used for comparisons among multiple samples.t
Thanks to the reviewer for pointing out the error. We rechecked the data methods and found this error. It was also found that this part of Materials and Methods is a duplicate content. There is already an explanation of data analysis in Figure 6, Figure 7, and Materials and Methods section 4.6. Therefore, this section was deleted after recognizing the error. Thank you very much for your helpful suggestion.
Round 2
Reviewer 1 Report
Comments and Suggestions for Authors
The resubmitted manuscript is significantly improved. I recommended that EiC accept the manuscript in its current form, after the necessary English check.
Comments on the Quality of English Languageminor check
Reviewer 3 Report
Comments and Suggestions for Authors
This manuscript has been improved a lot than the previous one.
Reviewer 4 Report
Comments and Suggestions for Authors
I have no other comment.